https://doi.org/10.1038/s42003-023-05586-4　　**OPEN**
# The structural basis for light acclimation in phycobilisome light harvesting systems systems in Porphyridium purpureum

Emma Joy Dodson[1,6], Jianfei Ma [2,6], Maayan Suissa Szlejf [3], Naama Maroudas-Sklare [4], Yossi Paltiel [4], Noam Adir [3], Shan Sun [2], Sen-Fang Sui [2,5✉] & Nir Keren [1✉]

Photosynthetic organisms adapt to changing light conditions by manipulating their light harvesting complexes. Biophysical, biochemical, physiological and genetic aspects of these processes are studied extensively. The structural basis for these studies is lacking. In this study we address this gap in knowledge by focusing on phycobilisomes (PBS), which are large structures found in cyanobacteria and red algae. In this study we focus on the phycobilisomes (PBS), which are large structures found in cyanobacteria and red algae. Specifically, we examine red algae (*Porphyridium purpureum*) grown under a low light intensity (LL) and a medium light intensity (ML). Using cryo-electron microscopy, we resolve the structure of ML-PBS and compare it to the LL-PBS structure. The ML-PBS is 13.6 MDa, while the LL-PBS is larger (14.7 MDa). The LL-PBS structure have a higher number of closely coupled chromophore pairs, potentially the source of the red shifted fluorescence emission from LL-PBS. Interestingly, these differences do not significantly affect fluorescence kinetics parameters. This indicates that PBS systems can maintain similar fluorescence quantum yields despite an increase in LL-PBS chromophore numbers. These findings provide a structural basis to the processes by which photosynthetic organisms adapt to changing light conditions.

[1] Department of Plant and Environmental Science, The Alexander Silberman Institute of Life Sciences, The Hebrew University in Jerusalem, Jerusalem, Israel. [2] State Key Laboratory of Membrane Biology, Beijing Frontier Research Center for Biological Structures, Beijing Advanced Innovation Center for Structural Biology, School of Life Sciences, Tsinghua University, Beijing, China. [3] Schulich Faculty of Chemistry, Technion – Israel Institute of Technology, 32000 Haifa, Israel. [4] Department of Applied Physics, The Hebrew University in Jerusalem, Jerusalem, Israel. [5] School of Life Sciences, Cryo-EM Center, Southern University of Science and Technology, Shenzhen, Guangdong, China. [6] These authors contributed equally: Emma Joy Dodson, Jianfei Ma. ✉email: suisf@mail. tsinghua.edu.cn; nir.ke@mail.huji.ac.il

Aquatic photosynthetic microorganisms exhibit a diverse array of light harvesting pigment-protein complexes (LHCs). One of the most versatile LHC groups are the phycobilisomes (PBS), used by cyanobacteria and red algae. Phycobilisomes present the most diverse optical properties, spanning the range from 500–620 nm[1]. LHCs transfer absorbed exciton energy–energy transfer (EET)—to photochemical reaction centers (RCs)[2–4]. This range is of particular importance in aquatic environments where red light is strongly attenuated. With increasing water column depth, light intensity is reduced and the spectra is progressively limited to the 480–490 nm range[5]. Absorption and Exciton energy transfer (EET) must therefore be regulated to respond to these changes in light quality (range of wavelengths to be absorbed) and intensity[6–12]. This control enables the homeostasis of charge separation and electron transfer rates in the photosynthetic apparatus[13,14]. From an evolutionary perspective, the electron transfer chains of all photosynthetic organisms are generally very similar while the LHCs differ in structure, cross-section for light absorption, and control mechanisms[15].

Cyanobacterial and red algae PBS structures are composed primarily of α and β protein subunits that bind together to form (αβ) monomers that self-assemble into higher levels of organization, forming trimers and hexamers. Hexamers are either stacked on top of each other to form rods or positioned adjacent to each other as core cylinders. Open tetrapyrrole chromophore molecules are covalently attached to conserved binding sites in both rods and cores by dedicated bilin lyases[16]. Linker proteins (LPs) are threaded through the hollow interiors of the rods to both stabilize assemblies and modify EET[17,18]. In the majority of studied cases, PBS rods radiate outward from the core that is situated on top of the thylakoid membrane, enabling EET to the RCs, especially to photosystem II (PSII)[19,20]. Additional PBS forms exist including core-less arrangements in *Acaryochloris marina*[21], and rod-less PBS in *Leptolyngbya sp. JSC-1* (when grown in far-red light)[6,7].

The close link between the structure of the PBS and its photosynthetic properties is well established. Its extensive spectral range results from the chromophore composition of its three protein classes: phycoerythrin (PE), which absorbs at ~493 nm and fluoresces at ~574 nm, phycocyanin (PC), which absorbs at 620 nm and fluoresces at 650 nm, and allophycocyanin (APC), which absorbs at 650 nm and fluoresces at ~660 nm[3]. The PBS pigment composition may vary and additional outlier cases have been reported[22,23]. The ratio between these classes determines the effective cross-section of the PBS. Based on energy transfer principles, the relative positions, distances and orientations of chromophores within the PBS directly affect energy transfer parameters[24].

According to classical physics approaches, the PBS is structurally designed for much less efficient EET than would be expected from a green lineage photosynthetic antenna. FRET theory predicts that EET efficiency will be determined by the number of pigments in a photosynthetic unit, the distance and orientation between pigments, and their spectral overlap[25]. In the PBS, the relatively large distance between chromophores—around 2 nm between neighboring chromophores within hexamers and even larger distances between chromophores in neighboring hexamers —dictates intermediate-strength energy coupling. This is expected to reduce EET efficiency, as compared to the more tightly coupled green lineage photosynthetic antenna[26]. In the PBS, the number of pigments that can contribute toward EET can exceed a thousand[27]. According to FRET random walk principles, EET efficiency should be inversely proportional to the number of pigments in a photosynthetic unit[28]. Work on light and shade-adapted plants demonstrated that this is the case for LHCII

systems[29]. In the case of PBS FRET random walk calculations determined that the length of multi-hexameric PBS rods extends beyond the maximum limit for maintaining energetic efficiency[26,30,31]. However, extending the rods in low light acclimated marine *Synechococcus* species was accompanied by improved energy transfer. In that case, a reorganization of the energy transfer path was suggested to allow multiple overlapping paths[32]. Indeed, it was shown in a computational simulation that EET in cyanobacteria and red algal PBS rods distributes evenly throughout the system, laterally as well as vertically, without relying on any single pathway along the length of the rod[33]. This challenges earlier conceptions regarding the energy transfer efficiency of intermediately coupled PBS systems.

The resolution of the complete *Porphyridium purpureum* (*P. purpureum*) red algae PBS structure[27] was an important step toward an understanding of PBS structure-function relations. *P. purpureum* is a laboratory strain that represents the group of free-living red algae on which extensive physiological and structural data exists[19,27,34]. In that study *P. purpureum* were grown under low light (LL) conditions and the structure of the PBS isolated from these cells (LL-PBS) was determined by cryo-electron microscopy (cryo-EM) to an overall resolution of 2.8 Å. Here, we present the structure of the PBS isolated from ML grown *P. purpureum* cells (ML-PBS), determined at 3.04 Å. We probed photo-physiological and biophysical properties of *P. purpureum* grown under either LL or ML light intensities. By comparing structures and photosynthetic performance under the two light conditions, aspects of the structural components responsible for biophysical differences were determined.

## Results and discussion

**Physiological responses to growth light intensity.** The two light intensities used in this study were chosen for acclimation of *P. purpureum* cultures –50 μmol photons m$^{-2}$ s$^{-1}$ (low light, LL) and 350 μmol photons m$^{-2}$ s$^{-1}$ (medium light, ML). LL represents the lowest light intensity under which this organism can grow in our cultivator setup. It also fits the light intensity used for growth in[27]. ML is midrange for growth in our setup and is considered only as ML as it is lower than the threshold for inducing extensive photo-inhibitory damage to reaction centers. Initial physiological differences could be observed by the naked eye within 3 days, as ML cultures became orange and LL cultures became purple (For further details see the methods section). This issue is dealt with quantitatively in Fig. 1. Pigment profiles in vivo were resolved by absorption spectroscopy, using an integrating sphere to reduce scattering. PBS chromophores absorb light at distinct wavelengths allowing for a quantitative comparison of LL (marked by blue lines and symbols, throughout the manuscript) and ML (marked by red lines and symbols) grown cells. Figure 1A indicates an overall four-fold increase in pigment content per cell in LL cultures as compared to ML cultures. After normalizing the spectra internally (Fig. 1B), the ratios between pigments indicate a significant increase in phycoerythrobilin band of the PE component (PEB 500–600 nm) in the LL cultures along with a smaller increase of phycocyanobilin in the PC component (PC ~ 620, APC absorption overlaps with the chlorophyll band).

The absorption spectra of extracted hydrophobic pigment is presented in Fig. 1C. It indicates a four-fold increase in chlorophyll content per cell in LL samples compared to ML samples, similar to what was observed in vivo. After internal normalization, the carotenoid band of the ML cultures (460–470 nm) shows a higher carotenoid /chlorophyll ratio than in the LL cultures (Fig. 1D). Since carotenoids are involved in photoprotective energy quenching[35,36], their higher abundance in ML cells is not surprising[37,38]. The differences in pigment

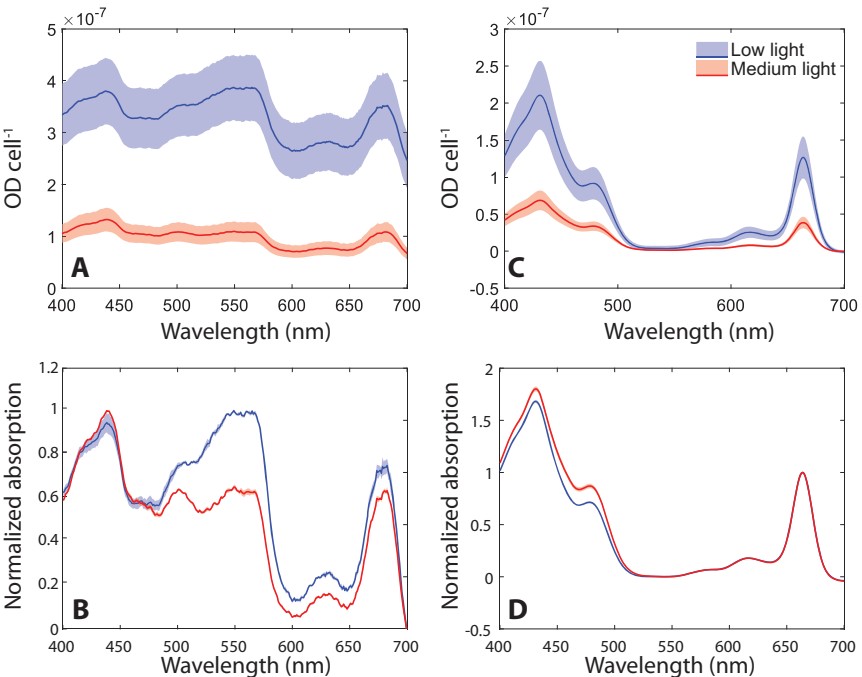

**Fig. 1 Pigment composition. A** In vivo total absorption spectra measurements, per cell; **B** Total absorption spectra from (**A**) internally normalized; **C** Absorption of extracted hydrophobic pigments, normalized per cell; **D** Membrane pigment absorption from (**C**) internally normalized at the 665 nm peak. Standard deviation- shaded area with $n = 3$.

**Table 1 Cryo-EM data collection, refinement and validation statistics.**

| | #1 name (EMDB-31393) (PDB 7EZX) |
|---|---|
| **Data collection and processing** | |
| Magnification | 130,000 |
| Voltage (kV) | 300 |
| Electron exposure (e-/Å$^2$) | 50 |
| Defocus range (um) | −1.3–2.3 |
| Pixel size (Å) | 1.066 |
| Symmetry imposed | C2 |
| Initial particle images (no.) | 288,782 |
| Final particle images (no.) | 87,399 |
| Map resolution (Å) | 3.03 |
| FSC threshold | 0.143 |
| Map resolution range (Å) | 2.64 ~ 6.74 |
| **Refinement** | |
| Initial model used (PDB code) | 6KGX |
| Map sharpening B factor (Å$^2$) | −62.05 |
| **Model composition** | |
| Non-hydrogen atoms | 1,021,204 |
| Protein residues | 126,474 |
| Ligands | 1602 |
| **B factors (Å$^2$)** | |
| Protein (mean) | 27.07 |
| Ligand (mean) | 38.41 |
| **R.m.s. deviations** | |
| Bond lengths (Å) | 0.010 |
| Bond angles (°) | 2.106 |
| **Validation** | |
| MolProbity score | 2.66 |
| Clashscore | 17.95 |
| Rotamer outliers (%) | 5.98 |
| **Ramachandran plot** | |
| Favored (%) | 95.44 |
| Allowed (%) | 4.42 |
| Disallowed (%) | 0.14 |

composition indicate structural rearrangements of both photosystems in the thylakoid membrane and the PBS light harvesting system associated with them. Here we will focus on the structural rearrangements of the PBS system.

**Structural data**. To explore the structural differences between ML-PBS and LL-PBS, we isolated the intact ML-PBSs, prepared cryo-EM samples, and collected cryo-EM data in a process similar to that used for LL-PBS (Fig. S1). The structure of ML-PBS was determined at an overall resolution of 3.04 Å, and the resolution of certain regions of the core even reached 2.2 Å (Supplementary Fig. S1 A–D and Table 1). To facilitate model building, individual local masks were applied to further improve the resolutions of some local regions as described in the methods (Fig. S1E). Finally, the atomic model was built based on the ML-PBS overall map and 14 local maps, in which the EM densities of pigments and side chains for most residues are clearly visible (Fig. S2, Supplementary Table S2). In total, we built 654 protein subunits including 480 PE subunits, 72 PC subunits, 46 APC subunits, and 56 linker proteins, and we assigned 1458 chromophores including 1298 PEB, 40 phycourobilin (PUB) and 120 phycocyanobilin (PCB) molecules (Supplementary Tables S2, S3). The molecular mass of ML-PBS is computed as 13.6 MDa, which is less than 14.7 MDa of LL-PBS (Supplementary Table S3). This calculation is consistent with the results obtained by sucrose gradient centrifugation, which indicates that ML-PBS has a smaller molecular mass than LL-PBS (Fig. S1). The overall architecture of ML-PBS is almost identical to that of LL-PBS: a pyramidal-shaped APC core is surrounded by fourteen peripheral rods arranged in a staggered fashion, and extra PE hexamers, individual αβ$^{PE}$ monomers and β$^{PE}$ subunits are scattered throughout the PBS (Fig. 2B–D).

When comparing the LL-PBS and ML-PBS structures, there is a clear, visible absence at the periphery of the complex. The second hexamers in rods F and G (and their symmetry mates– F', G') are missing from the ML structure (Fig. 3 and S1), which results in the reduction of 132 PEBs and 8 PUBs (8.8% of the total chromophore content) in ML-PBS compared with LL-PBS

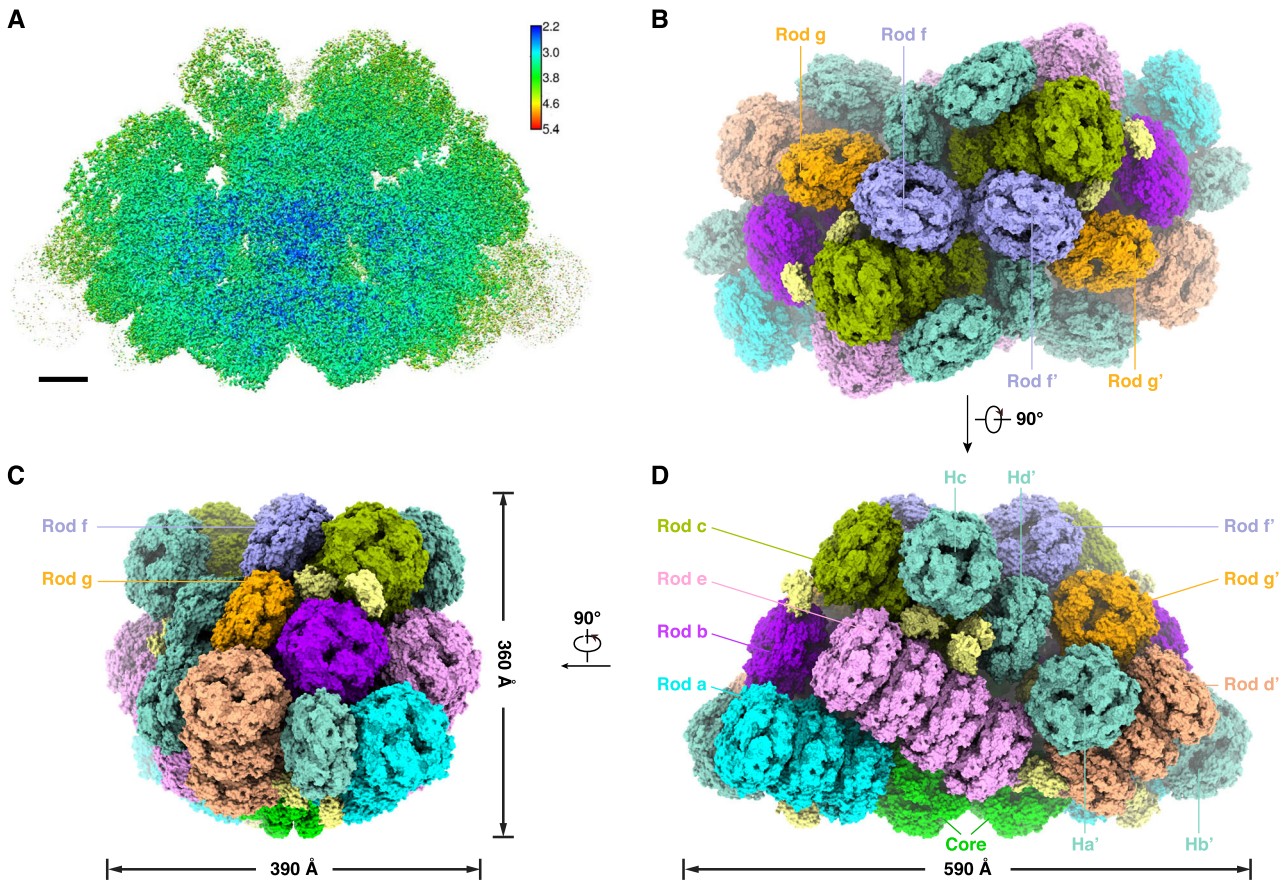

**Fig. 2 Overall structure of ML-PBS. A** Local resolution map of the ML-PBS from *P. purpureum*. The map was estimated with ResMap and generated in Chimera. **B–D** Structure of ML-PBS from three perpendicular views shown in surface representation. The rods (Rod a/a' to Rod g/g') are shown in different colors. The core, all extra PE hexamers (Ha/a' to Hd/d') are colored green and light sea green, respectively. All individual αβ^PE monomers and β^PE subunits are colored yellow.

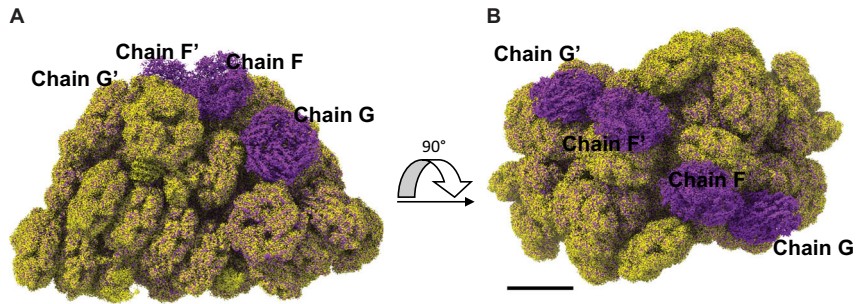

**Fig. 3 ML induced loss of terminal hexamers.** Cryo-EM density maps of LL PBS (purple) and ML PBS (yellow) show differences in protein structure and subunit organization. **A** side orientation shows the four hexamers missing in the ML structure. **B** 90 degrees rotation to a view from the top. The XY plane represents the direction of the thylakoid membrane, while the Z axis represents the direction of the stroma. The threshold for both maps is 0.01. Scale bar, 100 A.

(Supplementary Table 3). These results confirm our physiological characterization (Fig. 1), in which the ML cells are shown to have less phycoerythrin pigments per cell than LL. They also correspond to previous knowledge on light adaptation, where increasing the light intensity can result in loss of subunits and shorter rods in red algae[39] and marine *Synechococcus*[40,41]. The difference in the effective absorption cross-section between ML and LL PBS structures is in line with the difference in the excitation pressure between the two growth conditions. It should be mentioned that the hexamers' density is relatively poor even in the LL map, hence suggesting these hexamers to be flexible/

dynamic even in low light, which allows quick responses to changes in light conditions.

Higher resolution analysis reveals that the LL-PBS and ML-PBS structures differ in the distances between neighboring chromophore pairs. Closely coupled chromophore pairs in the *P. purpureum* PBS have already been identified in the LL structure[27]. While most interacting chromophores in the PBS structure are at least 2–3 nm apart, there are a few dozen pairs in the LL-PBS structure that are positioned significantly closer together. Figure 4 presents the distribution of chromophore pairs by center of mass-center of mass distance of up to 5 nm in the LL

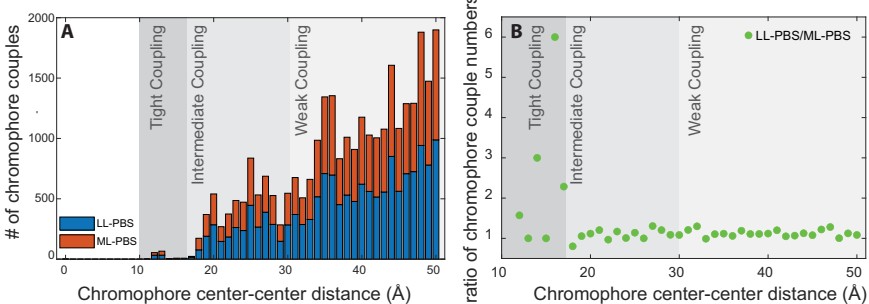

**Fig. 4 Distribution of chromophore-chromophore distances.** The stacked bar graph presents the distribution of chromophore distances. Chromophore distances up to 5 nm (center-center) apart are considered as relevant for efficient FRET. In the LL structure there are a total of 14,895 chromophore couples that meet this criterion. In the ML structure 13,334 couples meet this criterion. The histogram in (**A**) presents their distribution. Panel (**B**) is the ratio of LL-PBS/ML-PBS chromophore numbers. Only a small fraction of these pairs is tightly coupled— exhibiting distances of less than 1.5 nm. In total, we observed 74 pairs in LL (~0.5%) and 57 in ML (0.4%).

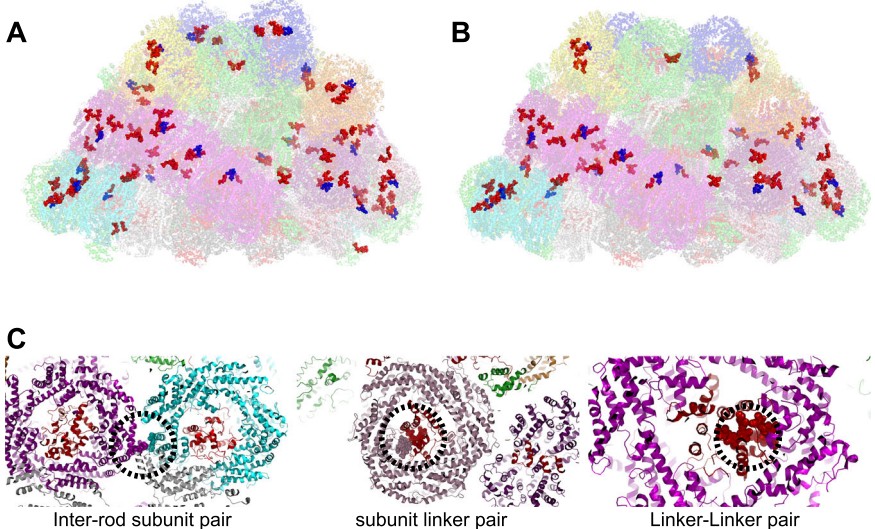

**Fig. 5 Closely coupled chromophore pairs.** The LL-PBS (**A**) and ML-PBS (**B**) structures and all chromophore pairs located less than 1.5 nm apart. The PBS protein structure is in transparent cartoon representation, and the closely coupled pairs are shown as spheres. All other chromophores are not shown. Red spheres—PEB, blue spheres—PUB. Panel (**C**) shows examples of an inter-rod subunit pair (left), a subunit-linker pair (middle), and a linker-linker pair (right). Hexamer subunit proteins and chromophores are colored in purple, pink, and cyan depending on which rod they are associated with, and linkers proteins and chromophores are colored in red.

and ML structures. The insert in Fig. 4 is a close-up of those chromophore pairs that are less than 1.5 nm apart, showing that a significant fraction of closely interacting chromophores is lost in the ML structure.

Phycoerythrobilin and phycourobilin chromophores are included in these close-coupled pairs, but not phycocyanobilin. There are 74 such pairs in the LL-PBS structure and 57 pairs in the ML-PBS structure. Many couples include one chromophore that is connected to linkers present in the central cavities of the hexamers/rods. Other couples are between chromophores connected only to the phycobiliprotein subunits of the rods. There are nine inter-rod pairs in the LL-PBS structure but only five such pairs in the ML-PBS structure. Notably, several closely coupled pairs are present in the four hexamers (blue and orange distal hexamers) that are lost in the acclimation from LL to ML (Fig. 5). The combination of close coupling and exclusive distribution on the periphery point to significant role for these pairs in PBS bioenergetics.

**Functional PBS responses to growth light intensity.** Considering the connection between structure and biophysical function,

further experiments were conducted to examine the impact of the structural differences on energy transfer and photosynthetic performance parameters. Room temperature fluorescence in the visible range was measured with excitation at 495 nm targeting PUB pigments (Fig. 6). The resulting emission spectra provide information on the fate of absorbed energy through the photosynthetic unit. Close examination of the PBS PE peak reveals a red shift of ~5 nm in the PE emission peak (575–580 nm) in LL as compared to ML cultures (Fig. 6 insert). This shift can be attributed to the difference in the number of closely coupled chromophore pairs. Delocalization of an excited state over a chromophore pair would result in red shifted absorption and emission[42].

Further fluorescence experiments were conducted to evaluate the PSII photochemical efficiency in LL and ML cells (Fig. 6). Measurements were performed in the presence or absence of DCMU, a competitive PSII electron transport inhibitor, that binds to the $Q_B$ site[43]. In the photosynthetic process, absorbed light is used for photochemistry, lost as heat, or emitted as fluorescence[44]. In the presence of DCMU, photochemistry is blocked. Assuming that changes in heat dissipation are negligible

under the low light intensities used in this measurement, all energy previously directed to photochemistry will be emitted as PSII fluorescence. Under these conditions, an apparent PSII photochemical yield, $QY_p$, can be determined by calculating the difference between the emission peak values of PSII in the presence or absence of DCMU (Eq. 1 in the methods section)[45]. The values for $QY_p$ were $0.65 \pm 0.01$ for the LL cultures and $0.55 \pm 0.01$ for the ML culture ($n = 3$, the difference is significant according to a two-sample unequal variance $T$ test at $p < 0.01$). These data indicate a higher PSII photochemical efficiency for the larger LL PBS structure. (It should be noted that PSI fluorescence is very low at RT)[44].

In Fig. 7a, the higher quantum yield of PSII photosynthetic units in LL cultures was validated using a different approach—pulse amplitude modulated (PAM) fluorometry. Here, a 450 nm

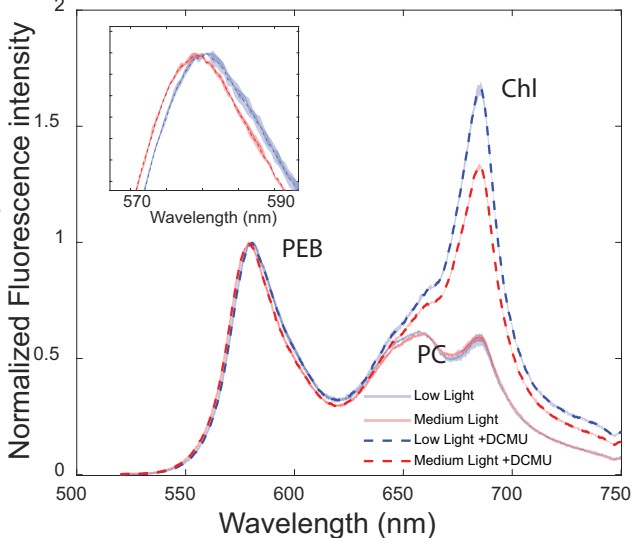

**Fig. 6 Room temperature fluorescence spectra.** Fluorescence emission (PUB excitation at 495 nm) in the presence and absence of DCMU, for ML and LL samples. Insert, close-up on the PE emission peak that shows a 3 nm shift in PEB emission. Standard deviation is presented as shaded area with $n = 3$.

LED actinic light, which preferentially excites the PBS PEB, as compared to chlorophyll, was used. Values for the maximal apparent PSII quantum yield (the PAM Fv/Fm parameter)[46] were consistently larger for LL cultures throughout the light curve. The induction of NPQ processes results in a decline in the apparent PSII quantum yield (YII) in both ML and LL cultures, as the actinic light intensity used during the measurement increases. Thus, the estimation of quantum yields was done by two different measurement approaches. While quantitatively it is hard to expect that they would fully agree with each other, they do agree with each other qualitatively.

Fluorescence lifetime (Time-Correlated Single-Photon Counting—TCSPC) measurements provide additional information on the dynamics of energy transfer through a pigment system (Fig. 7b). It is important to note that these measurements were performed in vivo and therefore demonstrate the effect of the consistent geometry between chromophores within individual rods, as well as that of the key inter-rod closely coupled pairs mentioned above. Therefore, the discrepancies in the measured parameters here and those measured in isolated hexamers in vitro (such as those reported by Li et al.[47]) are expected. A 495 nm pulse laser source was used. Detection was performed in three spectral windows corresponding with preferential PE, PC or Chl +APC emission. We based our conclusions not only on the lifetimes, but also on the spectral window in which they were measured, where a clear difference between PE and PC exists. As would be expected, the measured lifetimes increase with the increase in distance from the excited PE location. However, increasing the PBS antenna size did not result in the significant change in the lifetime. Overall, the decay lifetime constants are similar for ML and LL cultures (Fig. 7b insert and Supplementary Table 3). Therefore, we suggest that LL cultures retained similar energy transfer rates despite the increase in PBS size (and number of chromophores). It is important to note that this study is not sensitive in these measurements to the pre-equilibrium dynamics of PBS fluorescence (Supplementary Fig. S3), which would be very interesting but would require a streak-camera type setup to resolve[48,49]. Nevertheless, for improving characterization we have included the times of the maxima, which unfortunately are not short enough to resolve the questions in the tens of ps range (Supplementary Table 3).

In this study we focus on light intensity induced changes to PBS structures. It is clearly interesting to link these data to effects

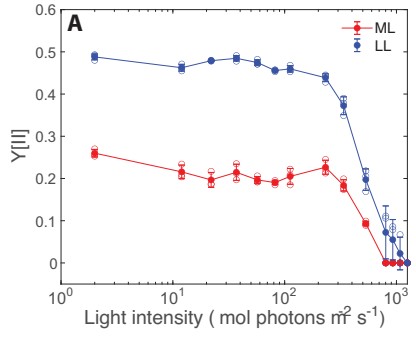

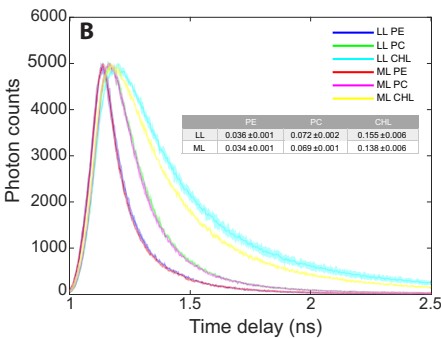

**Fig. 7 PAM and TCSPC fluorescence analyses. A** Quantum yield of LL and ML PSII measured using PAM fluorescence, with 450 nm actinic light exciting referentially PE. Detection was performed at wavelengths higher than 670 nm. The apparent PSII quantum yield parameter Y[II] was measured as a function of the actinic light intensity. $Y[II] = \frac{F'_m - F}{F'_m}$. Error bars represent standard deviation ($n = 3$, the differences between ML and LL values are significant according to a two-sample unequal variance $T$ test at $p < 0.01$ in the actinic light intensity range between dark adapted and 532 µmol photon m$^{-2}$ s$^{-1}$). **B** Fluorescence lifetime was studied by TCSPC. Excitation was at 495 nm and the data was collected in three detection windows: Preferential PE emission (PE 515–575 nm), Preferential PC emission (PC 650–670 nm) or preferential chlorophyll+APC emission (CHL 675–750 nm). The shaded areas represent standard deviation. The decay lifetime ($\tau$) was fitted with a single exponent for the PE and PC data and a double exponent for the chlorophyll data. Average $\tau$ values are presented in the inserted table ($n = 3$, the differences between ML and LL values are not significant according to a two-sample unequal variance $T$ test at $p < 0.01$). Additional data on the fitting procedure and data analysis is included in Supplementary Fig. S3.

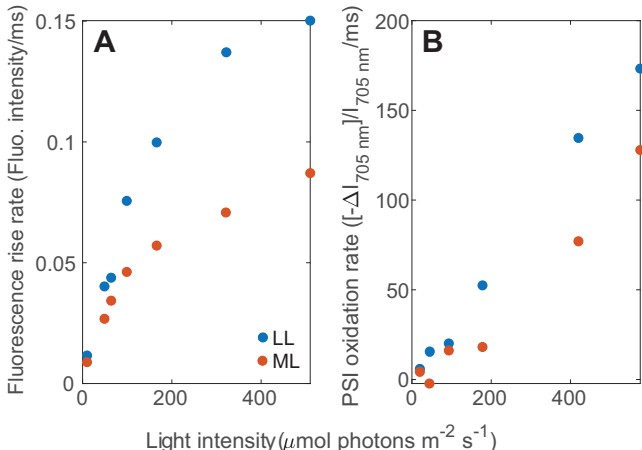

**Fig. 8 Effective PSII and PSI antenna size. A** Fluorescence rise rate showing initial rates of fluorescence emission by PSII following illumination in the presence of DCMU. **B** PSI oxidation rates showing initial rates of change in light absorption at 705 nm by PSI in the presence of DCMU. The experiment was repeated twice with comparable results.

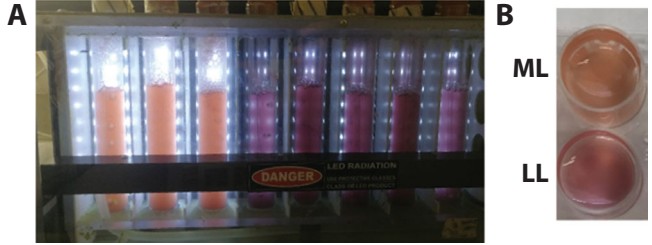

**Fig. 9 Growth conditions. A** The acclimated cultures in the photo-bioreactor. **B** A picture of culture samples displaying differential pigmentation.

on the photosystems to which they are coupled[50]. A quantitative estimation of photosystem content per cell is a major endeavor that is beyond the scope of this paper. Nevertheless, the chlorophyll to phycobilin ratio observed in Fig. 1 provides qualitative evidence of an increase in the relative phycobilisome content under LL conditions. This agrees with our previous studies, in which energy was shown to transfer through hexamer rods not in single, direct pathways, but rather by spreading laterally as well as vertically throughout the entire system[33].

Finally, we investigated the transfer of energy from the PBS to PSII and PSI by measuring the initial rate of fluorescence rise or P700 oxidation, respectively. These values serve as an indicator for PSII or PSI effective antenna size (Fig. 8). Using DCMU we blocked electron transfer out of PSII and into PSI. Under these conditions the rate of PSII fluorescence rise is a function of the effective antenna size[51]. Similarly, in the absence of re-reduction by electrons set in motion by PSII, the rate of PSI oxidation is a function of its effective antenna size. This oxidation is detected as absorption bleaching at 705 nm[52]. The fluorescence rise data indicates a larger functional PSII antenna size in LL cells as compared to ML cells. These results agree with the data presented in Figs. 8, 9. It is important to note that the structures (Figs. 2, 3) represent a physical absorption cross-section, while the data in Fig. 8 is an effective absorption cross-section. The physical cross-section is a basis for the effective cross-section but the two values may differ, as is well documented in the literature[53].

However, we also observed faster rates of $P_{700}$ oxidation in LL cells as compared to ML cells, indicating a larger functional PSI antenna size. The effect observed for the PSI effective cross-section can be the result of better coupling of PSI to the PBS system under LL conditions, or from indirect excitation spillover from PSII[20].

In this study, we observed three major differences when comparing ML-PBS and LL-PBS structures: (a) the number of closely coupled chromophore pairs (b) the overall number of peripheral hexamers and chromophores (c) the phycobilisome/ reaction center ratio. These structural features have significant implications for the photo-physiology of the algae. *P. purpureum* was isolated from shallow coastal waters and its photo-physiology evolved to meet the requirements of that environment[34]. Light propagation in coastal waters is strongly attenuated by the load of particulate and organic matter, resulting in large variability in both spectra and intensity. The ability to manipulate the spectral properties and the cross-section of the PBS may play an important role in acclimating to this environment. The acclimation properties identified here may extend to additional PBS system functioning under different light regimes. Pigmented linkers, which are responsible for several of the close chromophore pairs in the structure, are present in marine *Synechococcus* species[22,54]. These open ocean organisms also have the ability to manipulate their PBS cross-section[40].

However, the increase in cross-section does pose a bioenergetic problem. According to random walk FRET principles[55], exciton transfer efficiency should decrease with an increase in the chromophore network size. Nevertheless, fluorescence lifetime parameters, which are sensitive to changes in exciton transfer, are similar for the two different PBS systems. A possible explanation for the robustness of energy transfer may be found in the biophysical properties of the large intermediately coupled network of PBS chromophores. A recent simulation we performed for exciton transfer in these networks revealed that the ability to create multiple paths through the system makes it relatively immune to changes in number and orientation of individual components[33]. In essence, robustness is achieved with the price of lower exciton transfer efficiency—when comparing PBS to the tightly coupled chlorophyll a/b pigment systems of the green lineage[56].

Finally, we can consider the differences in PBS structure and function in terms of its contribution to the overall photosynthetic efficiency. Lower light acclimated algae exhibit higher photosynthetic quantum yield parameters. Acclimation to different light regimes involves changes that extend from light harvesting to photochemical reactions[57]. The bigger cross-section, spectral tuning ability and the robustness of energy transfer observed here can contribute towards the higher photochemical quantum yield recorded in the LL acclimated algae.

## Methods

**Growth conditions**. *Porphyridium purpureum* (UTEX Culture Collection, 2757) was grown in ASW[58] medium at 21 °C under low or medium light intensities, 50 and 350 μmol photons $m^{-2}$ $s^{-1}$, respectively, using glass tubes with a diameter of 3 cm. Cultures were grown in a photo-bioreactor (PSI Multi-Cultivator, Brno, Czech Republic) with constant bubbling (Fig. 9). Cultures were diluted every four days to maintain them in logarithmic phase. Cultures were given a week to acclimate in their respective light intensity environments before fluorescence and absorption measurements were taken.

**PBS preparation**. The culture conditions of *P. purpureum* cells used for PBS isolation are similar as above with small modifications. The cells were grown in Bold 1NV: Erdshreiber (1:1) half-seawater medium, bubbled with sterilizing filtered air at 22 °C,

under a 16 h:8 h light–dark cycle with a white-light flux of 350 μmol photons m$^{-2}$ s$^{-1}$. Preparation of PBS was performed as previously described[27]. The violet band located at the 1.0 mol/L sucrose gradient is the sample of intact PBSs used for single-particle analysis (Fig. S1).

**Cryo-EM sample preparation and data collection**. Cryo-EM grids were prepared with Vitrobot Mark IV (FEI Company) at 16 °C and 100% humidity. We used Cu-lacey carbon grids (200 mesh) with continuous carbon support film (Electron Microscopy Sciences) for cryo-EM sample preparation. After the grid was glow discharged for 30 s at low range, we added 1.5 μL aliquots of protein with concentration of 1.5 mg/mL to the grid and waited for 60 s, and then added 3.5 μL of 50 mM Tris buffer, pH 8.0 to the grid and mixed the sample quickly twice to reduce the salt concentration. The grid was then blotted for 3.5 s and plunged into liquid ethane cooled by liquid nitrogen.

Cryo-EM data were collected using a Titan Kiros Microscopy (FEI) operated at 300 KV and equipped with a GIF (Gatan) and a K2 Summit direct electron detector (Gatan). The defocus ranged from −1.3 to −2.3 μm. All cryo-EM images were recorded at a nominal magnification of ×130,000 under EFTEM Mode, corresponding to a calibrated pixel size of 1.08 Å. The K2 camera was operated in the super-resolution mode. Each movie stack has an accumulative dose of 50 electrons/Å$^2$ on the specimen. GIF was set to a slit width of 20 eV. The data were collected automatically using SerialEM software.

**Cryo-EM data analysis**. A total of 7631 micrographs were collected. In each movie stack, all of the frames were aligned and summed to correct the specimen drift and beam-induced motion using MotionCor2[59]. Micrograph screening, particle picking and normalization were performed by Relion 3.0 beta[60,61]. 6438 micrographs were selected for further processing. The contrast transfer function parameters of each micrograph were estimated by CTFFIND4[62] and all the 2D, 3D classification and refinement, CTF refinement were performed with Relion3.0 beta.

The workflow of cryo-EM data analysis is shown in Fig. S1E. Firstly, we performed particle auto-picking using 2D average as the template and then manually screen particles on all micrographs to eliminate the particle aggregation and ice contamination. 288,782 particles were selected for further processing. After 2 rounds of 2D classification, 286,910 particles were selected for the 3D classification. After 3D classification, 2 classes of particles (87,399) with reliable angle distribution were selected for the final reconstruction. Then we re-extracted particles from the dose-weighted micrographs and 3D refinement of these particles that yielded a map of 3.53 Å. By comparing this map with LL-PBS structure, we calibrated the pixel size from 1.08 to 1.066 Å/pixel. After re-evaluating the defocus of each particle by CtfRefine in Relion3.0 beta, we re-extracted particles with the pixel size of 1.066 Å/pixel and the box size of 580 pixel. Finally, we obtained a map at an overall resolution of 3.04 Å (Figs. S1 and S2). We applied individual local masks for the core region and some side regions, which resulted in improved quality of local maps (Fig. S1). The maps for the target regions were extracted from the overall map by CHIMERA[63], and the masks were created by RELION. All resolutions were estimated with the gold-standard Fourier shell correlation 0.143 criterion with high-resolution noise substitution (Fig. S1D). All the local resolution maps were calculated using ResMap.

**Model building and refinement**. The overall map together with 14 local maps were used to facilitate the model building process. We docked the model of LL-PBS (PDB ID 6KGX) into the ML-

PBS maps and most of the PBPs, linker proteins and pigments were fitted well. Then every residue was examined and manually adjusted to better fit the map in Coot. We also built L$_R$6, Linker3, CaRSP1 and CaRSP2 ab initio according to side chain information of bulky residues. The refinements of the individual model against the local map and the merged model against the overall map using phenix.real_space_refine in Phenix software[64] were performed as previously described[27]. The data collection, model refinement and validation statistics are presented in Supplementary Tables 1 and 2.

Model comparisons were performed using PyMOL[65]. Chromophore centers-of-mass were calculated by the PyMOL program and distances between chromophore pairs were measured using this data. Closely coupled pairs with distances below 1.5 nm were identified with these calculations and visualized using PyMOL. Both low light and medium light models can be found in the Protein Data Bank rcsb.org[66] (PDB 6KGX and PDB 7EZX, respectively). Map difference visualizations were generated using Chimera[63] and COOT[67] programs.

**Steady-state fluorescence and absorption spectra** were measured using an absolute PL quantum yield spectrometer (Quantaurus-QY—Hamamatsu photonics, Hamamatsu City, Japan). For the absorption spectra, samples were placed in an integrating sphere. Membrane pigments were extracted using 1:1 Methanol:Acetone according to the method described by Yoshida and coworkers[68]. Visible spectra were measured with A Carry-300Bio spectrometer.

Room temperature fluorescence was measured with 495 nm excitation aimed at the PUB component of the PBS[69]. The apparent PSII photochemical yield, $QY_p$, was calculated from the data as:

$$QY_p = \frac{F_{685nm}^{+DCMU} - F_{685nm}^{-DCMU}}{F_{685nm}^{+DCMU}} \tag{1}$$

**PAM fluorescence measurements** were used to determine the relative effective quantum yield of PSII under different actinic light intensities. The relative PSII quantum Yield values were determined using a Walz Imaging PAM (Effeltrich, Germany) light curve[70]. ML and LL samples were exposed to each light intensity in increasing steps of two minutes each.

**Time correlated single photon counting (TCSPC)** was used to quantify fluorescence lifetime. The measured lifetime is a harmonic sum of the three lifetime components corresponding to the three major pathways of energy dissipation: fluorescence ($\tau_f$), heat dissipation ($\tau_k$) and energy transfer ($\tau_{EET}$), so that $\frac{1}{\tau} = \frac{1}{\tau_f} + \frac{1}{\tau_k} + \frac{1}{\tau_{EET}}$ (equation 2). Cells were measured in vivo in a cuvette, using a reflection mode to minimize self-absorption. Measurements were conducted in a home-built TCSPC setup. Excitation was set at 495 nm, directed preferentially to the PUB, using Fianium WhiteLase SC-400 supercontinuum laser. Emission was collected from the different pigments using bandpass filters, and detected with MPD PD-100-CTE-FC photon counter and PicoHarp300, as described in Kolodny et al.[69]. To prevent unwanted second harmonics, the laser beam size is expanded to 2.5 mm diameter and the laser power is reduced from 0.91 W at source to ~40 nW, at point of incidence on the sample. This is achieved using a glass wedge, monochromator, and OD filter along the optical pathway. The setup was initially calibrated to the lowest intensity possible which still gives a reasonable Signal-to-Noise ratio to avoid non-linear effects. The time resolution of the system is 4 ps/channel. In order to sufficiently differentiate each photon pulse, the time between pulses is on the order of hundreds of ns, while the delay between the laser and the detector is set to 40 ps. The width of the instrument response function (IRF) is

95 ps for the 495 nm pulse (calculated by FWHM, See also Supplementary Fig. 3). Therefore, it is not sufficient to analyze lifetimes in the 10 ps region.

**Initial rise rate kinetics of PSI absorption and PSII fluorescence** were measured as a function of actinic light intensity using a Joliot-type spectrophotometer (JTS-10; Bio-Logic, Grenoble, France), sensitive in the microsecond time range. Measurements were conducted in the presence of 4 μM DCMU to prevent electron transfer between PSII and PSI.

**Statistics and reproducibility**. All biophysics measurements were performed with at least three biological repeats. Error bars were calculated as standard deviation values. TCSPC data was further analyzed using double exponential fits and $\chi^2$ statistical tests to evaluate the goodness of the fit (Supplementary Table 3). Further information on how the biophysical measurements and statistics were conducted can be found in the text and in the figure legends. Information on how those experiments lacking statistical analysis were conducted can also be found in the figure legends and in the text[71–74].

**Reporting summary**. Further information on research design is available in the Nature Portfolio Reporting Summary linked to this article.

## Data availability

The atomic coordinates have been deposited in the Protein Data Bank with the accession code 7EZX for ML-PBS. The EM maps for ML-PBS have been deposited in the Electron Microscopy Data Bank with the accession codes EMD-31393 for the overall map and EMD-31397 for core region, and EMD-31406 - EMD-31418 for the thirteen other local maps. Two un-sharpened PBS maps have been deposited in the Electron Microscopy Data Bank with the accession codes EMD-34529 for LL-PBS and EMD-34534 for ML-PBS. Data for the biophysical measurements can be found in Supplementary Data 1. All other data are available from the corresponding authors upon reasonable request. Detailed information, including all the raw data, can be found in Supplementary Data 1 (with all the data). It is arranged in Tabs matching the figure numbers.

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

## Acknowledgements

The Electron Microscopy data was collected at the Cryo-EM center of Southern University of Science and Technology in China. We thank the staff at the Tsinghua University Branch of the National Protein Science Facility (Beijing) for technical support on the High-Performance Computation platform. This work was supported by the National Natural Science Foundation of China (31861143048 awarded to S.-F.S., 32000848 awarded to J.M.) and by an NSFC-ISF Grant (2466/18 awarded to S.-F.S., N.K., Y.P., and N.A.). N.M.S. acknowledges the support of the Ministry of Energy, Israel, as part of the scholarship program for bachelor and graduate students in the fields of energy.

## Author contributions

E.J.D. and J.M. performed experiments, analyzed data, and wrote the manuscript, they are first co-authors of this paper; M.S.F. and N.M.S. analyzed data; Y.P., N.A., and M.S.S., analyzed data and wrote the manuscript; S.F.S. and N.K. conceived the project, supervised the work and wrote the manuscript, they are co-corresponding authors of the paper.

## Competing interests

The authors declare no competing interests.
