## [Peer Review File · Communications Biology]

Reviewers' comments:

Reviewer #1 (Remarks to the Author):

In the manuscript entitled "The structural basis for light acclimation in phycobilisome light harvesting systems", the authors examined the structures of phycobilisomes (PBSs) from *Porphyridium purpureum* cells grown under low-light (LL) and medium-light (ML) conditions. They revealed the ML-PBS structure with a resolution of 3.04 Å by the single-particle cryo-electron microscopy, and compared it to the LL-type structure. The LL-PBS is larger than the ML-PBS by 1.1 MDa, due to additional hexamers and chromophores. Although the antenna sizes are different between the LL- and ML-PBSs, fluorescence lifetimes did not show distinct differences, suggesting that the energy transfer rate be retained in the LL-PBS despite the larger antennas. The subject matter of this manuscript would be appropriate for *Communications Biology*. This reviewer suggests the following comments and questions.

1. Please provide information on how the lifetimes were analyzed from the fluorescence decay curves.
 - 1-1. Time resolution of the system and the width of instrumental response function need to be described in the text. Is the time resolution enough to analyze lifetimes in ten-picosecond region?
 - 1-2. Please show the fitting status of a single exponential decay to each observed fluorescence decay for phycobiliprotein. Although several energy transfer paths exist in PBS, do not the phycobiliproteins exhibit multi-exponential decays? The double exponential analysis for chlorophylls also seems simplified model.
 - 1-3. To show the validity of the authors' analysis, it would be better to draw the fluorescence decays in a logarithmic scale.
 - 1-4. Does not the fluorescence kinetics of phycocyanin exhibit a rise time of 34 to 36 ps, which corresponds to a lifetime of phycoerythrin? Chlorophyll should also exhibit (a) rise time(s).
2. The definition of "EET" is ambiguous (Introduction, line 3). "EET" might be an abbreviation of "Excitation Energy Transfer".
3. Introduction, line 6. It seems that two papers by Ho et al. reported the adaptation to far-red light. There might be better papers for the light quality and intensity.
4. Page 3, lines 4-6. The orientations between pigments are also important.
5. Page 3, line 12 from the bottom. "transfer" might be missing between "energy" and "efficiency".
6. Figure 1B. The 0 level should be corrected.
7. Figures 2 and 3. Please put a size scale on each figure.
8. Figure 6. The relative intensity of PEB is higher than that of PC, which seems to disagree with the fact that the lifetime of PEB is as half as that of PC. Does not PEB have (a) long-lifetime component(s) in addition to the 34-36-ps component?
9. Page 10, line 4 from the bottom. The light at 450 nm seems to excite chlorophyll also (Figure 1).
10. Figure 7, caption. "<670 nm" might be ">670 nm".
11. Page 18, line 17. "energy" should be replaced with "electron".
12. Suppl. Tab. 3. Is it possible that the PBSs exhibit (a) structure(s) between ML- and LL-PBSs under the intermediate light condition between ML and LL?

Reviewer #2 (Remarks to the Author):

The manuscript by Dodson et al. reports the structural basis for light acclimation in PBs in red microalga *Porphyridium purpureum*. In particular, the authors have characterized both cells' time resolved fluorescence lifetime, together with the JTS measurements on PSI and PSII antenna sizes. Overall, the manuscript is sound in terms of the experimental design and methodology applied. The statistical analysis of the data is well executed.

There are several minor issues that should be addressed before publishing.

Questions and comments:

1. Figure 1B, absorption is by no means negative;
 2. The values of QYp were deducted from Figure 6 by using the total area, I would suggest using the amplitudes at 680 nm, as it was named "an apparent PSII photochemical yield".
 3. For Figure 8, please specify the laser beam size and pulse-intensities used. And essentially, please include a laser power-dependent study to avoid any artifacts. btw, was DCMU+HA added too in these experiments? It is surprising to see that the ML and LL PE, PC curves overlapped so well.
 4. Why the antenna size of PSII is nearly doubled, see Figure 9A, while the number of chromophores is only 10% larger?
- Minor points:
Please pay attention to the significant digits, 0.28 ± 0.007 ; in Figure 7 caption, lied, probably yield?

Reviewer #3 (Remarks to the Author):

This is an impressive study of the structure and energy transfer properties of the phycobilisome antenna complexes in red algae grown under low and medium light intensities. The authors have done an excellent job of determining and describing the adaptations to changes in growth light intensity in these organisms.

The data are generally clear in the two sets of cells. I have only a few relatively minor issues to bring up. It is surprising to me that the quantum yield of photochemistry is higher in the low light cells compared to the medium light cells (Figs. 6 and 7). That seems counterintuitive and the two methods used to estimate the QY don't agree very well. If the quantum yield is higher for low light cells, then why can't the cells grow on even lower light intensities? Something doesn't seem right there.

The authors invoke changes in the PBS to RC stoichiometry to explain their results (beginning of Discussion), but don't report any direct measurements of that parameter. At a minimum, they need to address this issue head on and explain the results that lead to this conclusion.

Reviewer #4 (Remarks to the Author):

The authors investigated how the light harvesting complexes change in unicellular red algae *Porphyridium purpureum* grown under two light conditions, a low light intensity (LL) and a light intensity which is midrange for its photosynthetic capacity (ML), by using cryo-EM, total absorption spectra, fluorescence emission spectra, PSII photochemical yield and fluorescence lifetime measurements. The authors observed a decrease in the size of the PBS structure, number of closely coupled chromophore pairs, and the overall number of peripheral hexamers and chromophores in cryo-EM structures of ML-PBS compared to that of the LL-PBS. LL acclimation resulted in higher photosynthetic quantum yields than ML acclimation, and size of cross section, spectral tunability provides an important role in photo acclimation.

This is an interesting and important topic, which can be used as the basis for many other studies related to photo acclimation and excited energy transfer. Overall, the work is unique, as it combines findings of both structural biology and biophysical measurements related to how red algae acclimate to different light conditions by altering the structure of their light harvesting complexes. There are some matters which should be considered before final acceptance, listed below, but once dealt with, the work deserves publication.

Introduction

Page 2, Paragraph 1, line 2: One of the most versatile LHC groups are the phycobilisomes (PBS), used by cyanobacteria and red algae – Why is PBS one of the most versatile LHC groups?

Page 2, Paragraph 2, line 5: "Hexamers are either stacked on top of each other to form rods or positioned adjacent to each other as core cylinders." -This sentence should come before the open tetrapyrrole molecules (line 3) are mentioned.

Introduction could be further developed by adding a short paragraph on the changes in light availability in the ocean, how it would affect photosynthetic efficiency, and why it is important for the PBS structure to change under different light conditions. I believe it is important to discuss this as this paper is based on photo acclimation.

Page 2, Paragraph 3, Line 7: The authors use "absorbance cross-section", "absorption cross-section"(Page 7, paragraph 1, line 6) and "effective cross-section" (Page 12). Please specify whether this terminology refers to different things or not. If they all have the same meaning, please choose one term and be consistent throughout the manuscript.

Page 3, Paragraph 1, line 2: FRET theory predicts that EET efficiency will be determined by three main factors: the number of pigments in a photosynthetic unit, the distance between pigments, and their spectral overlap.
Please provide reference.

Page 3, Paragraph 1, Line 5: In the PBS, the relatively large distance between chromophores results in intermediate-strength energy coupling, which is expected to reduce EET efficiency.

What does "relatively large distances" mean in intermediate strength coupling? Is there any particular distance range used?

The authors mention terms such as intermediate-strength energy coupling, closely interacting chromophores (Page 7, Paragraph 2, line 2), closely coupled chromophores (Page 7, last line), significantly closer together (Page 7, Paragraph 2, line 5) throughout the paper. However, the method or references used to categorize chromophore distances under these terms have not been mentioned. I believe it is critical to define these parameters using the distance ranges that were considered in either the methodology (if it is unique to this paper) and/or in the introduction (if the categorization was used in a previous paper).

Page 3, Paragraph 1, Line 16-19: Indeed, it was shown that EET in cyanobacteria and red algal PBS rods distributes evenly throughout the system, laterally as well as vertically, without relying on any single pathway along the length of the rod (Dodson, 2022).

What technique was used here? Computational simulations? FRET, Fluorescence spectra?

Page 3, Paragraph 2, Line 6-7: Here, we present the structure of the PBS isolated from ML grown cells (ML-PBS), determined at 3.04 Å, which is comparable to that of LL-PBS.

Mention the organism of this study and why was this organism chosen in particular? What is the biological importance of this organism?

Results

Page 4, Paragraph 1, Line 3: LL represents the lowest light intensity under which this organism can grow in our cultivator setup.

The authors should explain why the LL and ML thresholds they chose are biologically important in the

real environment, not just in the cultivator setup, which is an artificial setting.

Page 4, Paragraph 1, Line 7: Initial physiological differences could be observed by the naked eye within 3 days, as ML cultures became orange and LL cultures became purple

The authors do not discuss how the color changes are related to photo acclimation. What pigments are responsible for these colors? Is there a significant increase in the respective pigments in LL and ML conditions? Can this be explained by absorption spectra measurements? What was the overall duration of the experiment and why was that duration chosen? These points should be expanded.

Page 5, Figure 1B: Is the intensity increase of the peak around 450 nm in ML cultures similar to that of Figure 1D? There is no explanation given by the authors.

Page 6, Line 1: 40 PUB and 120 PCB - Abbreviations should be defined.

Page 7, Paragraph 1, Lines 4-6: They also correspond to previous knowledge on light adaptation, where increasing the light intensity can result in loss of subunits and shorter rods (Ritz, 2000). This should go in the discussion. The reference cited is pretty outdated and should be replaced with more recent literature.

Page 7, Paragraph 1, Line 6-7: The difference in the absorption cross-section between ML and LL PBS structures in line with the difference in the excitation pressure between the two growth conditions. It is unclear what the authors referred to as the "absorption cross section" in the PBS structure. Please explain.

Page 6 and 7 Figures 2 and 3.

As this a cryo-EM paper, I believe the dimensions of the actual PBS structures can be included in terms of LL and ML cultures. Figure 1a and 1c in the paper cited below might be useful.

Zhang, J., Ma, J., Liu, D. et al. Structure of phycobilisome from the red alga *Griffithsia pacifica*. *Nature* 551, 57–63 (2017). <https://doi.org/10.1038/nature24278>

Although it is clear to some extent already, I suggest the authors to improve the resolution/color scheme of Figure 3 to clearly show the loss of terminal hexamers. Perhaps they could remove Figure 3 and add the LL and ML structures (just the xy plane which clearly shows all four missing hexamers in the ML structure) next to each other as E and F panels in Figure 2. Both structures could be gray, except for the hexamers on the LL that are missing in ML which could be highlighted with different colors (using the same graphics used for the structures displayed in Figure 2), to emphasize the parts that are missing in ML compared to LL.

Page 8, Figure 4, similar to comment 5, it would be more interesting to see if the authors can categorize the chromophore pairs as percentages of tight coupling, intermediate strength coupling, loose coupling, in addition to the figure given.

Page 11, Figure 7: Mention the sample size. What do the error bars represent in this Figure? Findings reported in Figures 6, 7 and 9 should be supported by statistical analyses which are missing throughout the manuscript. Are the differences they found statistically significant?

Page 11, Paragraph 2, line 9: number of complexes per RC: how was this concluded?

Discussion

There are several parts of the results section that read more like a discussion. Considering that the discussion is very scarce, I suggest the authors to combine their results and discussion sections into a single section and to significantly expand their discussion.

For instance, the authors begin the discussion by saying that when comparing ML-PBS and LL-PBS

structures there are differences in (a) the number of closely coupled chromophore pairs (b) the overall number of peripheral hexamers and chromophores (c) the phycobilisome/reaction center ratio. First of all, these variations need to be statistically tested to see whether they are significant. Secondly, they should explain why/how each of these changes (a-c) contribute to photo acclimation and whether these findings agree/disagree with previous studies.

Methods:

1. Page 15, Growth conditions, Paragraph 1, line 2: Please add reference to ASW media used. Why was 21 °C used as the temperature?
2. Page 17, spectroscopy, - how long were cultures maintained, and pre-acclimated to the LL, ML conditions before obtaining readings? Would long term acclimation influence the biophysical parameters measured in the paper?
3. No statistical analysis was mentioned in the methodology or results. It should be included.

Point by point response to the reviewers' comments.

(The main text file contains line numbers that we refer to in the point-by-point response. To further assist in identifying corrections in the text they are color coded, Reviewer#1-pink, Reviewer#2-green, Reviewer#3-cyan, Reviewer#4-gray)

Reviewer #1 (Remarks to the Author):

In the manuscript entitled “The structural basis for light acclimation in phycobilisome light harvesting systems”, the authors examined the structures of phycobilisomes (PBSs) from *Porphyridium purpureum* cells grown under low-light (LL) and medium-light (ML) conditions. They revealed the ML-PBS structure with a resolution of 3.04 Å by the single-particle cryo-electron microscopy, and compared it to the LL-type structure. The LL-PBS is larger than the ML-PBS by 1.1 MDa, due to additional hexamers and chromophores. Although the antenna sizes are different between the LL- and ML-PBSs, fluorescence lifetimes did not show distinct differences, suggesting that the energy transfer rate be retained in the LL-PBS despite the larger antennas. The subject matter of this manuscript would be appropriate for *Communications Biology*. This reviewer suggests the following comments and questions.

1. Please provide information on how the lifetimes were analyzed from the fluorescence decay curves.

1-1. Time resolution of the system and the width of instrumental response function need to be described in the text. Is the time resolution enough to analyze lifetimes in ten-picosecond region?

- Time resolution instrument response function is described in the Methods section (Lines 366-370). Time resolution is not sufficient to analyze data in the first 10 picoseconds of the measurement.
- The time resolution of the system is 4 ps. In order to sufficiently differentiate each photon pulse, the time between pulses is on the order of hundreds of ps, while the delay between the laser and the detector is set to 40 ps. Therefore, it is not sufficient to analyze lifetimes in the 10 ps region.
- The width of the instrument response function (IRF) is 95 ps for the 495 nm excitation (calculated by FWHM, see also new figure 8 and Supplemental Figure S3).

New Figure 8

New Supplemental Figure S3

1-2. Please show the fitting status of a single exponential decay to each observed fluorescence decay for phycobiliprotein. Although several energy transfer paths exist in PBS, do not the phycobiliproteins exhibit multi-exponential decays? The double exponential analysis for chlorophylls also seems simplified model.

The raw data is presented in Figure 8 (See above). The complete information on the fitting procedure is included in a new supplemental table (Table S4). All the PE/PC life-times were fitted to a single exponential function, while the Chlorophyll emission lifetimes were fitted to a double exponential function. Although it is true that a higher accuracy could be achieved by fitting to multiple exponents, the decision was made to fit a minimal number of exponents that achieve a low enough χ^2 (<16), since the objective was simply to compare different populations. All the raw data is provided in a supplemental excel file, enabling deeper analysis of the data in the future.

1-3. To show the validity of the authors' analysis, it would be better to draw the fluorescence decays in a logarithmic scale.

The Average data for the 3 repeats is presented in Log scale in Supplementary Figure S3.

1-4. Does not the fluorescence kinetics of phycocyanin exhibit a rise time of 34 to 36 ps, which corresponds to a lifetime of phycoerythrin? Chlorophyl should also exhibit (a) rise time(s).

It is important to note that we are not sensitive in these measurements to the pre- equilibrium dynamics of PBS fluorescence, which would be very interesting but would require a streak-camera type setup to resolve, such as that used by van Amerongen and others [Tian et al. (2012), Niedzwiedzki et al. (2022)]. Nevertheless, for improving characterization we have included the times of the maxima, which unfortunately are not short enough to resolve the questions in the tens of ps range. This explanation is included in the text (Lines 366-370).

2. The definition of "EET" is ambiguous (Introduction, line 3). "EET" might be an abbreviation of "Excitation Energy Transfer".

EET is defined lines 31-32.

3. Introduction, line 6. It seems that two papers by Ho et al. reported the adaptation to far-red light. There might be better papers for the light quality and intensity.

An additional reference to a current review was added (line 36).

4. Page 3, lines 4-6. The orientations between pigments are also important.

Thanks. The text was corrected (line 59).

5. Page 3, line 12 from the bottom. "transfer" might be missing between "energy" and "efficiency".

The text was corrected (line 80).

6. Figure 1B. The 0 level should be corrected.

Zero levels for the normalized data were corrected.

7. Figures 2 and 3. Please put a size scale on each figure.

Size scales were added.

New Figure 2

New Figure 3

8. Figure 6. The relative intensity of PEB is higher than that of PC, which seems to disagree with the fact that the lifetime of PEB is as half as that of PC. Does not PEB have (a) long-lifetime component(s) in addition to the 34-36-ps component?

The quantum efficiency of fluorescence emission is related to the lifetime, and the intensity is related to both the lifetime and the pigment concentration. PEB concentrations are much higher than PC concentrations. See, for example, Figure 1.

9. Page 10, line 4 from the bottom. The light at 450 nm seems to excite chlorophyll also (Figure 1).

The statement was revised to make it more accurate: Here, a 450 nm LED actinic light, which preferentially excites the PBS PEB, as compared to chlorophyll, was used. (Lines 193-194)

10. Figure 7, caption. "670 nm".

Corrected. (Line 606)

11. Page 18, line 17. "energy" should be replaced with "electron".

Corrected. (Line 374)

12. Suppl. Tab. 3. Is it possible that the PBSs exhibit (a) structure(s) between ML- and LL-PBSs under the intermediate light condition between ML and LL?

Thank you for this interesting question. We think it is possible that structures between ML-PBS and LL-PBS may exist, under the intermediate light condition between ML and LL, such as a structure in which the second hexamers in only one of rods F/F' and G/G' are missing. However, this will need to be addressed in research following up this paper.

Reviewer #2 (Remarks to the Author):

The manuscript by Dodson et al. reports the structural basis for light acclimation in PBS in red microalga *Porphyridium purpureum*. In particular, the authors have characterized both cells' time resolved fluorescence lifetime, together with the JTS measurements on PSI and PSII antenna sizes. Overall, the manuscript is sound in terms of the experimental design and methodology applied. The statistical analysis of the data is well executed.

There are several minor issues that should be addressed before publishing.

Questions and comments:

1. Figure 1B, absorption is by no means negative;

The figure was corrected (see also response to Reviewer #1 Comment 6).

2. The values of QYp were deducted from Figure 6 by using the total area, I would suggest using the amplitudes at 680 nm, as it was named “an apparent PSII photochemical yield”.

The calculation was corrected according to the reviewer’s suggestion (Lines 187-188).

3. For Figure 8, please specify the laser beam size and pulse-intensities used. And essentially, please include a laser power-dependent study to avoid any artifacts. btw, was DCMU+HA added too in these experiments? It is surprising to see that the ML and LL PE, PC curves overlapped so well.

To prevent unwanted second harmonics, the laser beam size is expanded to 2.5 mm diameter and the laser power is reduced from 0.91 W at source to approximately 40 nW at point of incidence on the sample. This is achieved using a glass wedge, monochromator, and OD filter along the optical pathway. The setup was initially calibrated to the lowest intensity possible which still gives a reasonable SNR to avoid non-linear effects. This information appears in the revised methods section (Lines 361-365).

DCMU and HA were not added to the experiments. Additional statistical information on the TCSPC data was added to the Supplemental Table S3 (See also Reviewer #1 comments 1-3 and 1-4). The overlap is indeed surprising; however, it correlates well with the other measurements performed here and in previous work on marine PBS energetics [Kolodny et al. (2021). Marine cyanobacteria tune energy transfer efficiency in their light-harvesting antennae by modifying pigment coupling. The FEBS Journal, 288(3), 980-994.].

4. Why the antenna size of PSII is nearly doubled, see Figure 9A, while the number of chromophores is only 10% larger?

The structure provides the physical absorption cross-section (Which is a measure for the probability of an absorption process), while in figure 9A we observe the functional absorption cross-section. These two values can differ, as is well documented in the literature (Gorbunov et al. (2001) A kinetic model of non-photochemical quenching in cyanobacteria, Biochimica et Biophysica Acta (BBA) - Bioenergetics, 12, 1591-1599). This is explained in the revised text (lines 230-234).

Minor points:

Please pay attention to the significant digits, 0.28 ± 0.007 ; in Figure 7 caption, lied, probably yield?

Corrected (Lines 187-188). Thanks.

Reviewer #3 (Remarks to the Author):

This is an impressive study of the structure and energy transfer properties of the phycobilisome antenna complexes in red algae grown under low and medium light intensities. The authors have done an excellent job of determining and describing the adaptations to changes in growth light intensity in these organisms.

The data are generally clear in the two sets of cells. I have only a few relatively minor issues to bring up. It is surprising to me that the quantum yield of photochemistry is higher in the low light cells compared to the medium light cells (Figs. 6 and 7). That seems counterintuitive and the two methods used to estimate the QY don’t agree very well. If the quantum yield is higher for low light cells, then why can’t the cells grow on even lower light intensities? Something doesn’t seem right there.

Correcting the QYp value calculations according to the suggestions of Reviewer 2 (comment 2) results in these values being in much better agreement with the PAM data in figure 7. The estimation of quantum yields is done by two different measurement approaches. While quantitatively it is hard to expect that they would agree with each other, they do agree with each other qualitatively (Lines 199-200).

Scanning the light intensity range for changes in PBS function, as suggested here, is an appealing idea for a physiological study. However, the need to connect the physiological data with structural analysis limited our options to two light intensities in which large enough biomass could be collected for PBS isolation and characterization.

The authors invoke changes in the PBS to RC stoichiometry to explain their results (beginning of Discussion), but don't report any direct measurements of that parameter. At a minimum, they need to address this issue head on and explain the results that lead to this conclusion.

The issue of PBS to RC stoichiometry was not addressed in the discussion. This was corrected. It is now mentioned in the following statement (Lines 215-219): In this study we focus on light intensity induced changes to PBS structures. It is clearly interesting to link these data to effects on the photosystems to which they are coupled. A quantitative estimation of photosystem content per cell is a major endeavor that is beyond the scope of this paper. Nevertheless, the chlorophyll to phycobilin ratio observed in Figure 1 provides qualitative evidence of an increase in the relative phycobilisome content under LL conditions

Reviewer #4 (Remarks to the Author):

The authors investigated how the light harvesting complexes change in unicellular red algae *Porphyridium purpureum* grown under two light conditions, a low light intensity (LL) and a light intensity which is midrange for its photosynthetic capacity (ML), by using cryo-EM, total absorption spectra, fluorescence emission spectra, PSII photochemical yield and fluorescence lifetime measurements. The authors observed a decrease in the size of the PBS structure, number of closely coupled chromophore pairs, and the overall number of peripheral hexamers and chromophores in cryo-EM structures of ML-PBS compared to that of the LL-PBS. LL acclimation resulted in higher photosynthetic quantum yields than ML acclimation, and size of cross section, spectral tunability provides an important role in photo acclimation.

This is an interesting and important topic, which can be used as the basis for many other studies related to photo acclimation and excited energy transfer. Overall, the work is unique, as it combines findings of both structural biology and biophysical measurements related to how red algae acclimate to different light conditions by altering the structure of their light harvesting complexes. There are some matters which should be considered before final acceptance, listed below, but once dealt with, the work deserves publication.

Introduction

Page 2, Paragraph 1, line 2: One of the most versatile LHC groups are the phycobilisomes (PBS), used by cyanobacteria and red algae – Why is PBS one of the most versatile LHC groups?

“Phycobilisomes present the most diverse range of optical properties, spanning the range from 500-620 nm.” This statement was added to the text (lines 30-31).

Page 2, Paragraph 2, line 5: “Hexamers are either stacked on top of each other to form rods or positioned adjacent to each other as core cylinders.” -This sentence should come before the open tetrapyrrole molecules (line 3) are mentioned.

The sentence order was corrected (lines 43-45).

Introduction could be further developed by adding a short paragraph on the changes in light availability in the ocean, how it would affect photosynthetic efficiency, and why it is important for the PBS structure to change under different light conditions. I believe it is important to discuss this as this paper is based on photo acclimation.

The light field in the ocean with respect to the water column depth is discussed in (lines 32-36) in the revised manuscript.

Page 2, Paragraph 3, Line 7: The authors use “absorbance cross-section”, “absorption cross-section”(Page 7, paragraph 1, line 6) and “effective cross-section” (Page 12). Please specify whether this terminology refers to different things or not. If they all have the same meaning, please choose one term and be consistent throughout the manuscript.

This was corrected and the terminology for absorption cross section and the effective absorption cross section is clarified (see extended response to Reviewer 2, comment 4).

Page 3, Paragraph 1, line 2: FRET theory predicts that EET efficiency will be determined by three main factors: the number of pigments in a photosynthetic unit, the distance between pigments, and their spectral overlap. Please provide reference.

The statement was corrected and a reference was added [Scholes, G. D. (2003). *Annu. Rev. phys. Chem.*, 54(1), 57-87] (lines 62-64).

Page 3, Paragraph 1, Line 5: In the PBS, the relatively large distance between chromophores results in intermediate-strength energy coupling, which is expected to reduce EET efficiency.

What does “relatively large distances” mean in intermediate strength coupling? Is there any particular distance range used?

This issue was dealt with extensively in a recent review paper by Keren and Paltiel, 2018. The text was extended to clarify the issue (lines 64-67).

The authors mention terms such as intermediate-strength energy coupling, closely interacting chromophores (Page 7, Paragraph 2, line 2), closely coupled chromophores (Page 7, last line), significantly closer together (Page 7, Paragraph 2, line 5) throughout the paper. However, the method or references used to categorize chromophore distances under these terms have not been mentioned. I believe it is critical to define these parameters using the distance ranges that were considered in either the methodology (if it is unique to this paper) and/or in the introduction (if the categorization was used in a previous paper).

The chromophore-chromophore distances which dictate in tight or intermediate coupling strengths are defined in introduction text (lines 64-67) and then again visually in Figure 4.

Page 3, Paragraph 1, Line 16-19: Indeed, it was shown that EET in cyanobacteria and red algal PBS rods distributes evenly throughout the system, laterally as well as vertically, without relying on any single pathway along the length of the rod (Dodson, 2022).

What technique was used here? Computational simulations? FRET, Fluorescence spectra?

This was a computational simulation study (corrected in the text, line 77).

Page 3, Paragraph 2, Line 6-7: Here, we present the structure of the PBS isolated from ML grown cells (ML-PBS), determined at 3.04 Å, which is comparable to that of LL-PBS.

Mention the organism of this study and why was this organism chosen in particular? What is the biological importance of this organism?

The following sentence was added to the revised text "*P. purpureum* is a laboratory strain that represents the group of free-living red algae on which extensive physiological and structural data exists." (Lines 83-85)

Results

Page 4, Paragraph 1, Line 3: LL represents the lowest light intensity under which this organism can grow in our cultivator setup.

The authors should explain why the LL and ML thresholds they chose are biologically important in the real environment, not just in the cultivator setup, which is an artificial setting.

The issue of selecting light intensities for these experiments and the limitations imposed by the need to generate enough biological material for structural analysis were brought up in our response to comment 12 by reviewer 1 and the first comment by Reviewer 3. The issue of natural light settings was addressed in the introduction following the first comment of Reviewer 4. More specifically to *P. Purpureum* - For additional information we added reference to work by Xie et al. (2021) "Difference in light use strategy in red alga between *Griffithsia pacifica* and *Porphyridium purpureum*. Scientific reports, 11(1), 14367." (Lines 85 and 245)

Page 4, Paragraph 1, Line 7: Initial physiological differences could be observed by the naked eye within 3 days, as ML cultures became orange and LL cultures became purple

The authors do not discuss how the color changes are related to photo acclimation. What pigments are responsible for these colors? Is there a significant increase in the respective pigments in LL and ML conditions? Can this be explained by absorption spectra measurements? What was the overall duration of the experiment and why was that duration chosen? These points should be expanded.

This point is dealt with quantitatively in figure 1. This statement was added to the text (Lines 218-219).

Page 5, Figure 1B: Is the intensity increase of the peak around 450 nm in ML cultures similar to that of Figure 1D? There is no explanation given by the authors.

We believe the reviewer is referring to the chlorophyll peak at wavelengths shorter than 450 nm. However, in the in vivo case it overlaps with phycoerythrobilin absorption, which are missing from

the methanol-extracted spectra. Therefore, a direct comparison of peak intensity is not straightforward.

Page 6, Line 1: 40 PUB and 120 PCB - Abbreviations should be defined.

Thanks. The abbreviations were defined (lines 131 and 132).

Page 7, Paragraph 1, Lines 4-6: They also correspond to previous knowledge on light adaptation, where increasing the light intensity can result in loss of subunits and shorter rods (Ritz, 2000). This should go in the discussion. The reference cited is pretty outdated and should be replaced with more recent literature.

Based on this reviewer's comments below, the structure of the paper was changed to results and discussion. Furthermore, newer reference where added (Kolodny et al. 2021, Bezalel-Hazony et al .2022; line 146).

Page 7, Paragraph 1, Line 6-7: The difference in the absorption cross-section between ML and LL PBS structures in line with the difference in the excitation pressure between the two growth conditions. It is unclear what the authors referred to as the "absorption cross section" in the PBS structure. Please explain.

These terms were changed to effective absorption cross-section, as per this reviewer's earlier comment (Line 147).

Page 6 and 7 Figures 2 and 3.

As this a cryo-EM paper, I believe the dimensions of the actual PBS structures can be included in terms of LL and ML cultures. Figure 1a and 1c in the paper cited below might be useful. Zhang, J., Ma, J., Liu, D. et al. Structure of phycobilisome from the red alga *Griffithsia pacifica*. Nature 551, 57–63 (2017). <https://doi.org/10.1038/nature24278>
A scale bar indicating the dimensions was added (See revised Figure 2 and further response to the next comment).

Although it is clear to some extent already, I suggest the authors to improve the resolution/color scheme of Figure 3 to clearly show the loss of terminal hexamers. Perhaps they could remove Figure 3 and add the LL and ML structures (just the xy plane which clearly shows all four missing hexamers in the ML structure) next to each other as E and F panels in Figure 2. Both structures could be gray, except for the hexamers on the LL that are missing in ML which could be highlighted with different colors (using the same graphics used for the structures displayed in Figure 2), to emphasize the parts that are missing in ML compared to LL.

Thank you for the response. We have added figure 3 with the scale bar and same graphics as in figure 2 - but suggest that the colors stay the same to clearly show the difference. If we change the threshold to make it more visible (0.01 instead of 0.017, first and second set of figures, accordingly). We added the following line to the text (at the end of the paragraph referring to figure 3) – "It should be mentioned that the hexamers' density is relatively poor even in the LL map, hence suggesting these hexamers to be flexible/dynamic even in low light, which allows quick response to changes in light conditions" (Lines 148-150).

Page 8, Figure 4, similar to comment 5, it would be more interesting to see if the authors can

categorize the chromophore pairs as percentages of tight coupling, intermediate strength coupling, loose coupling, in addition to the figure given.

We believe that the ratio plot provides a good handle for this information. To make the presentation clearer this plot was moved from insert to a separate panel and the coupling strength regions, as a function of chromophore-chromophore distance are indicated. See new Figure 4.

Page 11, Figure 7: Mention the sample size. What do the error bars represent in this Figure? Findings reported in Figures 6, 7 and 9 should be supported by statistical analyses which are missing throughout the manuscript. Are the differences they found statistically significant?

Error bars represent standards deviation. This appears in the text now. Statistical tests were performed for the data in figures 6 and 7 (Lines 188-189). For Figure 9 two independent repeats were performed giving qualitatively similar differences. This information is specified in the figure legend (Lines 608-610).

Page 11, Paragraph 2, line 9: number of complexes per RC: how was this concluded?

See response to Reviewer #3, final comment.

Discussion

There are several parts of the results section that read more like a discussion. Considering that the discussion is very scarce, I suggest the authors to combine their results and discussion sections into a single section and to significantly expand their discussion.

For instance, the authors begin the discussion by saying that when comparing ML-PBS and LL-PBS structures there are differences in (a) the number of closely coupled chromophore pairs (b) the overall number of peripheral hexamers and chromophores (c) the phycobilisome/reaction center ratio. First of all, these variations need to be statistically tested to see whether they are significant. Secondly, they should explain why/how each of these changes (a-c) contribute to photo acclimation and whether these findings agree/disagree with previous studies.

Results and discussion are combined in the revised manuscript. As for the issue of variation in chromophore coupling in the two structures:

The difference in all three parameters (a,b and c) are real numbers, based on real measurements, of thousands of molecules. The numbers themselves do not require statistics to bear out their validity. However, looking at figure 5 from our paper and Fig. 1 in the cryo-tomography pictures shown in the

Li et al. eLife 2021 paper, it appears that there are two “significantly different” positions of removal of close contacted bilin chromophore pairs: at the “bottom” close to the membrane (and RC chlorophylls) or near the “top” of the PBS. This second position (“top”) is actually adjacent to the “bottom” position in the nearby PBS complexes (as seen by tomography) – i.e., it will have the same result as there will be less lateral EET interaction from these chromophores directly to the nearby complexes at the position near the RCs. In our structure, we show that the number of close contacts at the “bottom” position, near the RC, goes down from 4 to 0 – that is certainly significant.

Methods:

1. Page 15, Growth conditions, Paragraph 1, line 2: Please add reference to ASW media used. Why was 21 °C used as the temperature?

This organism grows well at 21 deg, similar to many other marine organisms. A reference for ASW was added (Line 274).

2. Page 17, spectroscopy, - how long were cultures maintained, and pre-acclimated to the LL, ML conditions before obtaining readings? Would long term acclimation influence the biophysical parameters measured in the paper?

Longer acclimation to high light results in dense cultures in which the cells experience lower effective light intensities (See for example, Kolodny et al. 2022). Acclimation time was added to the text in the beginning of the Methods section (Lines 278).

3. No statistical analysis was mentioned in the methodology or results. It should be included.

Statistical methods were employed and in the revised manuscript as detailed above (Lines 188-189,608-610).

Reviewers' comments:

Reviewer #1 (Remarks to the Author):

The authors well answered this reviewer's small questions and comments, but there still remain serious concerns on the time-resolved measurement and its following analysis.

Li et al. reported that phycoerythrin of *Porphyridium purpureum* exhibits the 8-ps and 60-ps energy transfers (Chinese Journal of Physics, 66 (2020) 24–35), although the present manuscript by Dodson suggests only a ~30-ps decay in the phycoerythrin fluorescence region. In addition, if phycocyanin does not exhibit a ~30-ps rise, it is difficult to assign this time constant to the main energy-transfer route between phycoerythrin to phycocyanin, and the main transfer might be occurring in the earlier time region. Therefore, the time resolution of time-resolved measurements in the present manuscript does not seem enough to detect differences between energy-transfer kinetics in phycobilisomes (PBSs) from *Porphyridium purpureum* cells grown under the low-light (LL) condition and those in PBSs from the cells grown under the medium-light (ML) condition. To bring the authors' conclusion "LL cultures retained similar energy transfer rates despite the increase in PBS size (and number of chromophore) (lines 208–209)", another measurement with higher time resolution is required.

Line 366. "ps" might be "ps/channel"

Line 367. "ps" might be "ns".

References #6 and #8 do not seem to focus on changes in energy transfer under different light conditions. Many researchers have examined the responses of energy-transfer processes to the light quality and quantity. It seems better to cite other earlier publications.

Table S4.

The values of χ^2 are so high that it does not seem possible to claim that the authors' analysis is accurate.

The value of a_1 for LL chl 1 is 1.000. Is the contribution of the second component 0?

Even though the authors state that their system is not sufficient to analyze lifetimes even in the 10 ps region (lines 369-370), the lifetime values are written down to the order of picoseconds.

Reviewer #2 (Remarks to the Author):

All the comments I made were properly addressed by the authors

Reviewer #3 (Remarks to the Author):

The authors have adequately responded to the comments by the reviewers. I support the manuscript for publication in its current form.

Reviewer #4 (Remarks to the Author):

Reviewer Comments

Overall, the authors have done a great job in addressing the questions and suggestions provided. The new figures, 2 & 3 look much more clearer and addition of proper dimensions make a noticeable change. I believe the manuscript is now at a standard to which it can be published in Communications

Biology. My further comments are as given below.

1. Line 283: Did authors mean 16 h: 8 h light-dark cycle? Instead of 6 h: 8h?
2. Point by point addressing reviewer comments - Page 8, sentence below is cut short.
"If we change the threshold to make it more visible (0.01 instead of 0.017, first and second set of figures, accordingly."

Response in blue

Reviewer 1:

Li et al. reported that phycoerythrin of *Porphyridium purpureum* exhibits the 8-ps and 60-ps energy transfers (Chinese Journal of Physics, 66 (2020) 24–35), although the present manuscript by Dodson suggests only a ~30-ps decay in the phycoerythrin fluorescence region. In addition, if phycocyanin does not exhibit a ~30-ps rise, it is difficult to assign this time constant to the main energy-transfer route between phycoerythrin to phycocyanin, and the main transfer might be occurring in the earlier time region. Therefore, the time resolution of time-resolved measurements in the present manuscript does not seem enough to detect differences between energy-transfer kinetics in phycobilisomes (PBSs) from *Porphyridium purpureum* cells grown under the low-light (LL) condition and those in PBSs from the cells grown under the medium-light (ML) condition. To bring the authors' conclusion "LL cultures retained similar energy transfer rates despite the increase in PBS size (and number of chromophore) (lines 208–209)", another measurement with higher time resolution is required.

In Li et al., *in vitro* measurements are performed on analytical grade B-PE hexamers that have been extracted from *P. purpureum* cells, i.e. no longer exist in the context of intact antennas. Our measurements were performed *in vivo* on intact antenna systems. It is therefore hard to expect the energy transfer parameters to be the same, as the rod structure is built in such a way as to facilitate interaction between inter-hexamer PEBs (keeping them far enough apart to prevent extreme oscillation but not so far apart as to prevent all interaction), thus affecting their energetics. Beyond this, there are several inter-rod pairs of closely coupled chromophores in key locations in both the LL and ML structures, as discussed in our article (see Figure 5). Interestingly, there are almost twice as many in the LL structure as in the ML structure (Two examples are presented in the figure below). The closeness between these chromophores will certainly cause coupling between them and have an effect on their energetics (a detailed analysis of the energetic effects of chromophore coupling can be found in Dodson et al. [2022] Journal of the Royal Society Interface).

Two examples of closely coupled neighbors in the PBS structure.

Top: Chromophores on the Ra rod (cyan) and its inter-rod neighbor on the Rb rod (purple) in the LL structure. The distance between their centers of mass was calculated as 12.0 Å.

Bottom: Chromophores in the Re rod (magenta) and Rc (yellow) rods in the LL structure.

See also Figure 5 in the paper.

For a complete list of inter-rod chromophore pairs in the LL structure, please see the table below:

Chain	Residue	Rod	Chain	Residue	Rod
P7	202	Ra	QI	202	Rb
Q2	202	Rb2	PF	202	Ra2
C3	202	Rg	G4	202	Ha
F9	202	Rf2	FJ	202	Rf
G1	202	Ha	CD	202	Rg2
OB	202	Rc	qG	203	Re
K4	202	Ha2	KE	203	Re2
K1	202	Ha	KG	203	Re
O6	202	Rc2	qE	203	Re2

In an *in vitro* context, the inter-hexamer and inter-rod spacing along with the controlled coupling in key areas of the antenna system will no longer be maintained and the PEB energetics will certainly change, and with that the lifetime parameters. Therefore, a direct comparison between *in vivo* and *in vitro* PE hexamers is not straight forward. This is now mentioned in the text.

To make these points clear, we have added the following statements to the text (lines 203-207 and 209-210):

- It is important to note that these measurements were performed *in vivo* and therefore demonstrate the effect of the consistent geometry between chromophores within rods, as well as that of the key inter-rod closely coupled pairs mentioned above. Therefore, discrepancies in the measured parameters here and those measured isolated hexamers *in vitro* (such as those reported by Li et al.....) are expected.
- We base our conclusions not only on the lifetimes, but also on the spectral window in which they were measured, where a clear difference between PE and PC exists.

Furthermore, we agree with the reviewer that better time resolution will provide information on events that are faster than the decay lifetime, measured after thermal equilibrium, that are determined and analyzed in this work. Higher time resolution measurements are beyond the scope of experimentation methods in our labs at this moment in time.

Line 366. "ps" might be "ps/channel"

Yes, this has been changed to ps/channel (now line 372).

Line 367. "ps" might be "ns".

Yes, this has been changed to hundreds of ns (now line 373).

References #6 and #8 do not seem to focus on changes in energy transfer under different light conditions. Many researchers have examined the responses of energy-transfer processes to the light quality and quantity. It seems better to cite other earlier publications.

The following classical works in the field have been added:

Glazer, Alexander N. "Structure and molecular organization of the photosynthetic accessory pigments of cyanobacteria and red algae." *Molecular and cellular biochemistry* 18 (1977): 125-140.

Grossman, A. R. (2003). A molecular understanding of complementary chromatic adaptation. *Photosynthesis research*, 76, 207-215.

Bryant, D. A. (1982). Phycoerythrocyanin and phycoerythrin: properties and occurrence in cyanobacteria. *Microbiology*, 128(4), 835-844.

Tandeau de Marsac, N. (1977). Occurrence and nature of chromatic adaptation in cyanobacteria. *Journal of bacteriology*, 130(1), 82-91.

Table S4.

The values of χ^2 are so high that it does not seem possible to claim that the authors' analysis is accurate.

The fits were performed in a manner that targeted a specific χ^2 threshold. Were they to be performed for multiple exponents, we would have arrived at lower χ^2 values, but this is just analogous to adding extra variables to an equation to improve the fit and we didn't see the added value in doing so. The single and double exponent fits were deemed adequate for our purposes.

The value of a1 for LL chl 1 is 1.000. Is the contribution of the second component 0?

To the accuracy of 3 decimal places, the contribution is 0, yes. The value given by the fitting software was 0.9995.

Even though the authors state that their system is not sufficient to analyze lifetimes even in the 10 ps region (lines 369-370), the lifetime values are written down to the order of picoseconds.

Correct, the fitting values are given to a higher degree of accuracy to make the data calculation more transparent. As such, we do not make any claims as to the differences observed as a result, even though the software output gives a small enough standard deviation. Our only claim is the similarity between ML and LL samples.

Reviewer 2:

Line 283: Did authors mean 16 h: 8 h light-dark cycle? Instead of 6 h: 8h?

Thank you, this has been corrected (now line 289).

Point by point addressing reviewer comments - Page 8, sentence below is cut short. "If we change the threshold to make it more visible (0.01 instead of 0.017, first and second set of figures, accordingly."

This response has been changed to:

Thank you for the response. We have added figure 3 with the scale bar and same graphics as in figure 2 - but suggest that the colors stay the same to clearly show the difference.

We added the following line to the text (at the end of the paragraph referring to figure 3) – “It should be mentioned that the hexamers' density is relatively poor even in the LL map, hence suggesting these hexamers to be flexible/dynamic even in low light, which allows quick response to changes in light conditions” (Lines 148-150).

REVIEWERS' COMMENTS:

Reviewer #1 (Remarks to the Author):

The followings are suggestions for improving the manuscript:

If the readers are familiar with measurements and analyses of excited state dynamics, they will see the large values of χ^2 , and wonder if there are problems with measurements and/or analyses. When the χ^2 values were not improved through the analysis, it would be better to modify the experimental system or procedure. In this manuscript, the lifetimes of PE were analyzed to be 0.03-0.04 ns, which are less than half of the IRF width (95 ps). Especially in such cases, it is very important to obtain a sufficiently low χ^2 value to determine the lifetime.

To avoid confusion for readers, the lifetime values in Supplemental Table 4a need to be modified to be consistent with the description in the experimental section (lines 375-376). According to the description "it is not sufficient to analyze lifetimes in the 10 ps region", the significant figures are up to the order of 100 ps or 10 ps, not up to 1 ps. Thus, for example, "0.034 ns" should be "0.03 ns". Based on the above conditions, it should be better at the present stage to use "it was suggested" rather than "we can conclude" (lines 214-215).

Response in blue:

Reviewer #1 (Remarks to the Author):

The followings are suggestions for improving the manuscript:

If the readers are familiar with measurements and analyses of excited state dynamics, they will see the large values of χ^2 , and wonder if there are problems with measurements and/or analyses. When the χ^2 values were not improved through the analysis, it would be better to modify the experimental system or procedure. In this manuscript, the lifetimes of PE were analyzed to be 0.03-0.04 ns, which are less than half of the IRF width (95 ps). Especially in such cases, it is very important to obtain a sufficiently low χ^2 value to determine the lifetime.

To avoid confusion for readers, the lifetime values in Supplemental Table 4a need to be modified to be consistent with the description in the experimental section (lines 375-376). According to the description "it is not sufficient to analyze lifetimes in the 10 ps region", the significant figures are up to the order of 100 ps or 10 ps, not up to 1 ps. Thus, for example, "0.034 ns" should be "0.03 ns".

Based on the above conditions, it should be better at the present stage to use "it was suggested" rather than "we can conclude" (lines 214-215).

We acknowledge Reviewer #1's concern and have changed the statement in line 214 from "we can conclude" to "we suggest".